# Fabrication of Silk Fibroin Fluorescent Nanofibers via Electrospinning

**DOI:** 10.3390/polym11060986

**Published:** 2019-06-04

**Authors:** Liaoliao Pang, Jinfa Ming, Fukui Pan, Xin Ning

**Affiliations:** Industrial Research Institute of Nonwovens and Technical Textiles, College of Textiles & Clothing, Qingdao University, Qingdao 266071, China; ll_pism@163.com (L.P.); xning@qdu.edu.cn (X.N.)

**Keywords:** fluorescence, silk fibroin, nanofiber, electrospinning

## Abstract

Fluorescent silk fibroin nanofibers were fabricated via electrospinning method with three kinds of fluorescent dyes. Electrospun fluorescent nanofibers showed smooth surfaces and average diameters of 873 ± 135 nm, 835 ± 195 nm, and 925 ± 205 nm, respectively, for silk fibroin-fluorescein sodium, silk fibroin-rhodamine B, and silk fibroin-acridine orange nanofibers containing 2.0 wt% fluorescent dyes. At the same time, the secondary structure of silk fibroin in fluorescent nanofibers was predominantly amorphous conformation without influence by adding different concentrations of fluorescent dyes, as characterized by Fourier transform infrared spectroscopy and X-ray diffraction. Thermal degradation behavior of fluorescent silk fibroin nanofibers with a dramatic decrease in weight residue was observed at around 250 °C. The fluorescence effect of fluorescent silk fibroin nanofibers was changed by changing the concentration of different fluorescent dyes. These fluorescent nanofibers may make promising textile materials for large scale application.

## 1. Introduction

Nanomaterials exhibit excellent characteristics, such as large surface area and high porosity, compared with other known forms of materials [1]. With the development of nanotechnology, a number of processing techniques, including melt fibrillation, nanolithography, template synthesis, self-assembly, interfacial polymerization, and electrospinning, have been used to prepare nanofiber materials in recent years [2]. The main advantages of electrospinning among other techniques is it being a convenient, low-cost, and efficient method to fabricate uniform and continuous nanofibers [3]. These nanofibers have been used for biomedicine, filtration and separation, protective clothing, sensors, and energy devices [4,5,6].

Fluorescent electrospun nanofibers are attracting increasing interest for their potential exploitation in textiles, photonic applications, and so on. For example, Li et al. showed fluorescent CdTe quantum dots were dispersed into poly(vinyl alcohol) (PVA) nanofibers by electrospinning, generating a highly fluorescent 1D material with dramatically high aspect ratios [7]. Wang et al. prepared pyrene/polystyrene/tetrabutylammonium hexafluorophosphate (pyrene/PS/TBAH) fluorescent nanofibrous films for the detection of ultra-trace nitro explosive vapors and buried explosives by the naked eye under UV excitation [8]. Wen et al. reported that poly[(*N*-isopropylacrylamide)-*co*-(*N*-hydroxymethyl acrylamide)-*co*-(4-rhodamine hydrazonomethyl-3-hydroxy-phenyl methacrylate)] (PNNR) copolymers were synthesized by free radical copolymerization of three monomers. In acidic environments, PNNR copolymers showed highly selective and sensitive recognition and displayed “ON-OFF” fluorescence toward Cu^2+^, both in solution and in solid state [9]. Hu et al. studied 4-chloro-7-nitrobenzo oxadiazole/o-phenylendiamine (NBD-OPD) and 1,8-naphthalimide/o-phenylendiamine (RB-OPD) embedded electrospun polymer fibers, which displayed distinct color and fluorescence changes upon exposure to phosgene, even in the solid state [10]. Wang et al. prepared polyvinylpyrrolidone Janus nanofiber membranes via electrospinning method. These Janus nanofiber membranes exhibited excellent magnetic performance and high fluorescent performance due to the unique structure [11]. In addition, fluorescent nanofibers were also produced via introducing a small amount of inorganic quantum dots into polymer fibers [2].

Silk fibroin (SF), a natural fibrous protein produced by Bombyx mori, has been used for textile and biomedical applications [12,13]. To suit a wide range of applications, SF has been integrated with various materials or been chemically modified. For example, fluorescent SF materials are produced by silkworms with a fluorescence-dye-containing diet or directly by dyeing with fluorescence dyes [14]. Kusurkar et al. used a “water glass” based strategy to isolate the fluorophores from silk cocoons [15]. Kim et al. described a novel method of preparing a fluorescent SF solution used for biomedical applications [16]. Min et al. reported a novel combination of silk biopolymer and optically active organic dyes resulting in versatile fluorescent silk nanofibers [17].

As is commonly known, fluorescent nanofibers have been the subject of continued interest. There have been several reports on the biomedical and biotechnological applications using fluorescent silk fibroin. However, there is almost no application in the field of textiles, such as fluorescent strips and fluorescent icons for apparel. In this paper, we report herein the demonstration of electrospun fluorescent silk nanofibers with stable fluorescence performance for textile applications. The functionalities of fluorescent SF nanofibers depend on organic dyes doped in the silk solution. High-quality fluorescent nanofibers with spatial homogeneities in diameter and fluorescent emissions were obtained for three different fluorescent dyes. Furthermore, the effect of SF and fluorescent dye concentration on the fiber morphology and size distribution are systematically studied. The thermal stability and fluorescence properties of these composite fibers are investigated.

## 2. Materials and Methods

### 2.1. Materials

Bombyx mori silkworm cocoons were purchased from Shandong province, China. Fluorescein sodium (FS) and rhodamine B (RB) were obtained from Sinopharm Chemical Reagent Co. Ltd. (Shanghai, China). Acridine orange (AO) was purchased from Shanghai Macklin Biochemical Co. Ltd. Other reagents (sodium chloride, formic acid, calcium chloride, etc.) were also bought from Sinopharm Chemical Reagent Co. Ltd. (Shanghai, China), and directly used without purification.

### 2.2. Electrospinning for Generation of Fluorescent Silk Nanofibers

Bombyx mori silk fibroins were prepared according to published procedures [18]. Degummed silk was dissolved in formic acid-CaCl_2_ (FA-CaCl_2_) solvent at room temperature. This solution was evenly poured into culture dishes and dried for film formation, and then, films were soaked in deionized water to remove Ca^2+^ ions. After removing Ca^2+^ ions, films were dried at room temperature.

Dried films were re-dissolved in formic acid (FA) solution and stirred at room temperature to obtain the base solution. Three types of fluorescent agents, including RB, AO, and FS were used, and their chemical formulas are shown in Appendix A. The concentration of SF and fluorescent agents in FA solution are shown in Table 1. After stirring at room temperature, the spinning solution was prepared. The prepared SF bio-ink was loaded into the syringe and mounted on the electrospinning machine.

Fluorescent SF nanofibers were produced and the preparation parameters were as follows. The spinning voltage was 15 kV. The tip-to-collector distance was fixed to 15 cm. The spinning solutions were loaded into a 10 mL syringe, and the inner diameter of the metal needle was 0.9 mm. The flow rate of the solution was 0.2 mL h^−1^. After electrospinning, fluorescent SF nanofibers were dried at room temperature overnight to dry off the remaining solvent.

### 2.3. Characterization

**Scanning electron microscope (SEM)**: The surface morphology of fluorescent nanofibers was observed by SEM (S4800, Hitachi, Chiyoda-ku, Japan). All samples were sputter coated with gold prior to imaging. The average diameter of fluorescent SF nanofibers was obtained by analyzing SEM images using image J analysis software, and the value of each sample was calculated by a diameter of one hundred nanofibers.

**Structural analysis**: The secondary structures of fluorescent SF nanofibers were analyzed by FTIR on Nicolet5700 (Thermo Nicolet Company, MA, USA) in absorbance mode. For each measurement, each spectrum was obtained by the performance of 32 scans with the wave number ranging from 400 to 4000 cm^−1^ with a resolution of 4 cm^−1^. Meanwhile, in order to investigate the crystalline structure of the fluorescent SF nanofibers, X-ray diffraction experiments were measured on X Pert-Pro MPD (PANalytical, Almelo, The Netherlands) in transmittance mode. The incident beam wavelength was 0.154 nm. The intensity was finally corrected for changes in the incident beam intensity, sample absorption, and background.

**Thermal analysis**: The thermal stability of fluorescent SF nanofibers was characterized using SDT Q600 (TA Company, Boston, MA, USA) under nitrogen gas flow of 40 mL min^−1^. Samples were heated from room temperature to 600 °C, and the heating rate was 10 °C min^−1^.

**Fluorescence analysis**: The fluorescence emission spectra of fluorescent nanofiber samples were carried out using Hitachi F-4600 Spectrofluorometer (Hitachi High-Technologies Corporation, Hitachinaka, Japan). During the test, every sample was folded into two layers and laid on a solid support of a fluorescence spectrophotometer. The operating voltage was set to 250 V. The slit width of excitation light and emitted light was 5 nm. The excitation wavelength was chosen to be 450 nm and the emission wavelength range was between 450 and 800 nm.

### 2.4. Statistical Analysis

All values were expressed as mean ± standard deviation. Statistical differences were determined by a Mann-Whitney *U* test (Independent *t* test, SPSS14.0).

## 3. Result and Discussion

### 3.1. Morphology of Electrospun Fluorescent SF Nanofibers

For fluorescent dyes involving nanofiber electrospinning, electrospinnability of nanofibers played a critical role. In this study, fluorescent SF nanofibers with different fluorescent dyes could be readily electrospun into fibrous form (Figure 1) owing to the excellent electrospinnability of the SF solution system.

Figure 1 showed the morphology of fluorescent SF nanofibers with different FS contents. Pure SF nanofibers exhibited circular cross-sections with smooth surfaces (Figure 1f). For an image analysis, the average diameter of pure SF nanofibers was measured as 390 ± 45 nm. At the same time, electrospun SF-FS nanofibers containing 1.0 wt% FS and 6.0 wt% SF were observed to be continuous and geometrically fairly uniform, with a diameter of 419 ± 53 nm (Figure 1a), which was larger than the diameter of pure SF nanofibers. With increasing SF solution concentration ranging from 6.0 wt% to 10.0 wt%, the diameter of fluorescent SF-FS nanofibers increased from 419 ± 53 nm to 683 ± 126 nm and its distribution became significantly broader (Figure 1b,c). Furthermore, at the concentration of SF 10.0 wt%, the average diameter of fluorescent SF-FS nanofibers was 1200 ± 261 nm and 873 ± 135 nm, containing 0.5 wt% and 2.0 wt% FS, respectively (Appendix A). This phenomenon was attributed to the dissolution degree of SF and SF concentration in electrospinning solution.

Figure 2 depicted the morphology of fluorescent SF-RB nanofibers with different SF and RB concentrations. For RB concentration 1.0 wt%, electrospun SF-RB nanofibers have smooth surfaces and their diameters were 150 ± 40 nm, 299 ± 66 nm, and 677 ± 132 nm (Appendix A), respectively, with increasing SF concentration ranging from 6.0 wt% to 10.0 wt% (Figure 2a–c). At the same time, when the concentration of RB was 0.5 wt% and 2.0 wt% in nanofibers, the average diameter of fluorescent SF-RB nanofibers was 779 ± 135 nm and 835 ± 195 nm for 10.0 wt% SF concentration (Figure 2d,e). The diameter change was greatly affected by the concentration of RB in the electrospinning solution.

Figure 3 showed the morphology of fluorescent SF-AO nanofibers with different SF and AO concentrations. For AO concentration of 1.0 wt%, electrospun SF-AO nanofibers have smooth surfaces and their diameters were 190 ± 40 nm, 1042 ± 277 nm, and 691 ± 146 nm (Appendix A), respectively, with increasing SF concentration ranging from 6.0 wt% to 10.0 wt% (Figure 3a–c). When the concentration of AO was 0.5 wt% and 2.0 wt% in nanofibers, the average diameter of fluorescent SF-AO nanofibers was 804 ± 148 nm and 925 ± 205 nm for 10.0 wt% SF concentration (Figure 3d,e). The diameter change was also confirmed and affected by the concentration of fluorescent dyes in the electrospinning solution.

In addition, fluorescent nanofibers were produced by electrospinning method, and influenced by three kinds of fluorescent dyes (FS, RB, and AO). The morphology of these fluorescent nanofibers exhibited a circular cross-section with smooth surface (Figure 1e, Figure 2e, and Figure 3e). At the same time, the average diameters were 873 ± 135 nm, 835 ± 195 nm, and 925 ± 205 nm, respectively, for SF-FS, SF-RB, and SF-AO nanofibers. The results from different fluorescent dyes had no significant effect on the morphology and diameters of fluorescent nanofibers.

### 3.2. Structural Characteristics

FTIR spectrum was carried out to confirm the functional groups in the fluorescent SF nanofibers (Figure 4 and Figure 5). Figure 4 reveals the secondary structure of SF in fluorescent nanofibers. The peaks of pure SF nanofibers were characterized at 1655 cm^−1^ (amide I), 1536 cm^−1^ (amide II), and 1239 cm^−1^ (amide III), corresponding to amorphous conformation (Figure 4Ab). When different fluorescent dyes were added with 1.0 wt% concentration, and the concentration of SF increased from 6.0 wt% to 10.0 wt%, the peak positions of the characteristic bands of SF were not shifted (Figure 4A–C), demonstrating mainly the amorphous conformation in fluorescent nanofibers. It also showed different fluorescent dyes were not significantly affected by the secondary structure of SF.

Figure 5 depicts FTIR spectra of fluorescent SF nanofibers with different concentration of FS. The main characteristic peaks of pure SF nanofibers were located at 1655 cm^−1^, 1536 cm^−1^, and 1239 cm^−1^, corresponding to the amorphous structure (Figure 5a). When the concentration of FS was 0.5 wt%, the peaks at 1646 cm^−1^, 1528 cm^−1^, and 1243 cm^−1^ were observed, corresponding to amorphous conformation (Figure 5b). For FS concentration increasing up to 2.0 wt%, the peak positions of the characteristic bands of SF were not shifted, demonstrating mainly the amorphous conformation in fluorescent nanofibers (Figure 5c,d). At the same time, different concentrations of FS were not significantly affected by the secondary structure of SF.

The crystal structure of fluorescent SF nanofibers containing different types and concentration of fluorescent dyes was also analyzed by XRD (Figure 6). In previous research studies three SF conformations had been identified by XRD: random coil, α-form (silk I, type II β-turn), and β-form (silk II, anti-parallel β-pleated sheet) [19]. Figure 6A shows XRD data of fluorescent SF nanofibers with different concentrations of FS. When the concentration of FS was 0.5 wt%, XRD data of fluorescent SF-FS nanofibers did not obviously exhibit the main characteristic peaks of FS materials. The characteristic peaks of SF were located at 9.5°, 20.8°, and 23.7°, corresponding to amorphous conformation (Figure 6Aa). Figure 6Ab depicted the diffraction peaks at 2θ values of 9.5°, 20.7°, and 22.8°, showing that the structure of SF in fluorescent nanofibers was predominantly an amorphous structure. However, the concentration of FS increased to 2.0 wt% (Figure 6Ac), and all characteristic peaks were consistent with the peaks of other samples (Figure 6Aa,b). This implied different concentration of FS added in fluorescent nanofibers were not affected by the secondary structure of SF. In addition, fluorescent dyes (RB, AO) were added in fluorescent nanofibers through electrospinning method. Figure 6B,C also revealed different concentrations of RB and AO addition in fluorescent nanofibers were not affected by the secondary structure of SF.

### 3.3. Thermal Stability Analysis

From thermogravimetric curves results, a weight loss of about 7.0 wt% around 100 °C was detected in pure SF nanofibers, which was ascribed to the evaporation of water (Figure 7Aa). As the temperature was increased further, the weight residue started to decrease sharply at 250 °C, due to the thermal degradation of silk fibroin. When fluorescent SF nanofibers contained 1.0 wt% FS, a similar water content as at about 6.0 wt% was also observed around 100 °C; moreover, similar thermal degradation behavior with a dramatic decrease in weight residue was observed at around 250 °C (Figure 7Ab). When FS concentration was increased up to 2.0 wt% in fluorescent SF nanofibers, the mass loss of the sample was similar to pure SF nanofibers (Figure 7Ac). Furthermore, Figure 7B,C depicts the thermogravimetic curves of fluorescent SF nanofibers with different concentrations of RB and AO. The results also show the thermogravimetric trend of fluorescent SF-RB and SF-AO nanofibers was similar to fluorescent SF-FS nanofibers. This phenomenon implied the crystal structure of SF was amorphous conformation, and was not transformed under addition of the different types and concentrations of fluorescent dyes.

### 3.4. Fluorescent Performance Analysis

Fluorescent SF nanofibers were electrospun by adding different types and concentrations of fluorescent dyes. These nanofibers were placed in a dark place and irradiated with an ultraviolet lamp to observe their fluorescence effect (Figure 8). The result exhibited the fluorescence effect of fluorescent nanofibers was changed by changing the concentration of different fluorescent dyes. At the same time, the fluorescence emission spectra of fluorescent SF nanofibers were studied with an excitation spectrum at 450 nm over the emission spectra in the range of 450-800 nm. Excitation and emission spectra of fluorescent nanofibers with different concentrations of SF and fluorescent dyes were analyzed (Figure 9). Figure 9A1 shows the fluorescence intensities of all fluorescent SF-FS nanofibers at 540 nm, without affecting the concentration of SF. For 0.5 wt% FS concentration, the location of fluorescence emission was 530 nm. Within an increase of FS concentration from 0.5 wt% to 2.0 wt%, the location was shifted right from 530 nm to 535 nm (Figure 9A2). The fluorescence emission shift was correlated to the fluorescence effect of fluorescent SF nanofibers irradiated by UV light (Figure 8). The fluorescence effect of fluorescent SF nanofibers was changed by changing the concentration of FS. Hence, the fluorescence emission shift was also affected by the concentration variations of FS.

For fluorescent SF-RB nanofibers, the maximal emission was located at 615 nm (Figure 9B1a,b). Figure 9B1c shows the fluorescence emissions of samples were shifted left from 615 nm to 605 nm. Within increasing RB concentration, the fluorescence emission shifted from 615 nm to 620 nm (Figure 9B2). Moreover, the fluorescent effect of fluorescent SF-AO nanofibers was similar to fluorescent SF-RB nanofibers (Figure 9C1,C2). This phenomenon for fluorescence emission shift change was also attributed to the fluorescence effect of nanofibers, which were affected by the concentrations of RB and AO, respectively.

## 4. Conclusions

Fluorescent SF nanofibers were prepared via electrospinning method with different fluorescent dyes. Fluorescent SF nanofibers containing 2.0 wt% fluorescent dyes could be electrospun into the continuous fibrous structure, although fluorescent SF nanofibers showed larger diameters and broader diameter distribution than pure SF nanofibers. SEM results showed the average diameters were 873 ± 135 nm, 835 ± 195 nm, and 925 ± 205 nm, respectively, for SF-FS, SF-RB, and SF-AO nanofibers containing 2.0 wt% fluorescent dyes. Different fluorescent dyes had no significant effect on the morphology and diameters of fluorescent nanofibers. At the same time, the secondary structures of fluorescent SF nanofibers were characterized by FTIR, XRD, and thermal analysis. Comparing with pure SF nanofibers, the structure of SF in fluorescent nanofibers was predominantly amorphous conformation, without being influenced by adding different concentrations of fluorescent dyes. The crystal structure was confirmed by FTIR and XRD. Thermal analysis results showed thermal degradation behavior of fluorescent SF nanofibers with a dramatic decrease in weight residue was observed at around 250 °C. Furthermore, the fluorescence effect of fluorescent SF nanofibers was changed by changing the concentrations of different fluorescent dyes. These fluorescent nanofibers may make promising textile materials for large scale application.

## Figures and Tables

**Figure 1 polymers-11-00986-f001:**
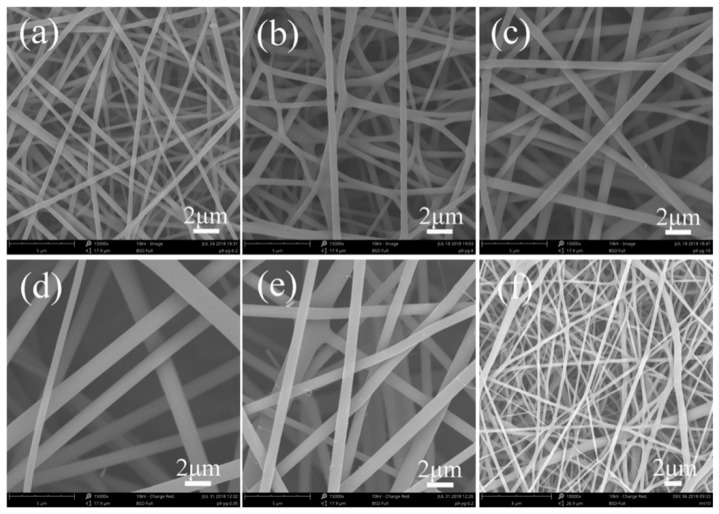
SEM images of fluorescent SF-FS nanofibers with different SF and FS concentrations. The concentration of SF was (**a**) 6.0 wt%, (**b**) 8.0 wt%, and (**c**–**e**) 10.0 wt%. The concentration of FS was (**a**–**c**) 1.0 wt%, (**d**) 0.5 wt%, and (**e**) 2.0 wt%, respectively. (**f**) Pure SF nanofibers with 6.0 wt% SF concentration.

**Figure 2 polymers-11-00986-f002:**
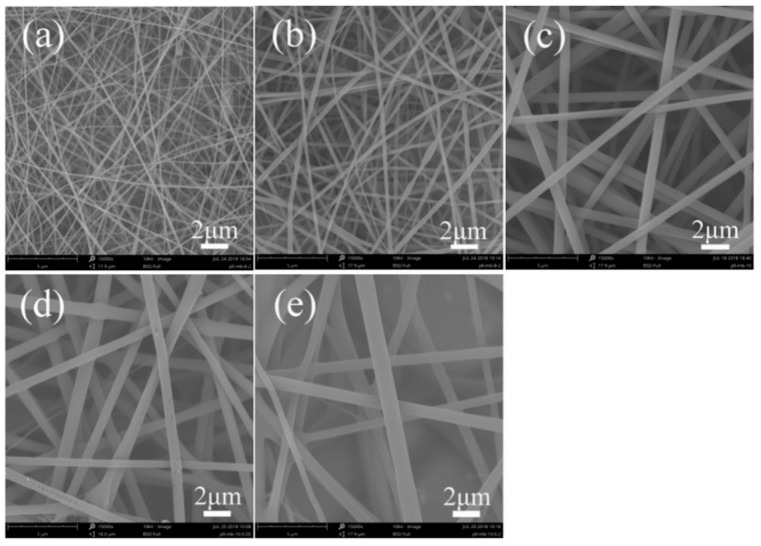
SEM images of fluorescent SF-RB nanofibers with different SF and RB concentration. The concentration of SF was (**a**) 6.0 wt%, (**b**) 8.0 wt%, (**c**–**e**) 10.0 wt%. The concentration of RB was (**a**–**c**) 1.0 wt%, (**d**) 0.5 wt%, and (**e**) 2.0 wt%, respectively.

**Figure 3 polymers-11-00986-f003:**
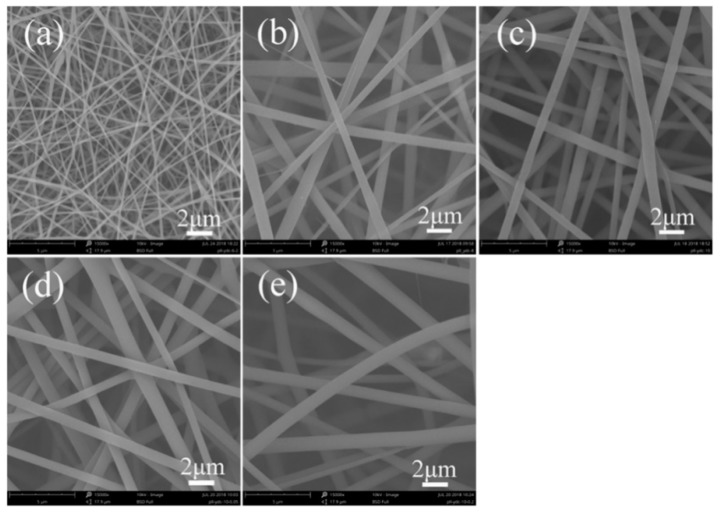
SEM images of fluorescent SF-AO nanofibers with different SF and AO concentration. The concentration of SF was (**a**) 6.0 wt%, (**b**) 8.0 wt%, (**c**–**e**) 10.0 wt%. The concentration of AO was (**a**–**c**) 1.0 wt%, (**d**) 0.5 wt%, and (**e**) 2.0 wt%, respectively.

**Figure 4 polymers-11-00986-f004:**
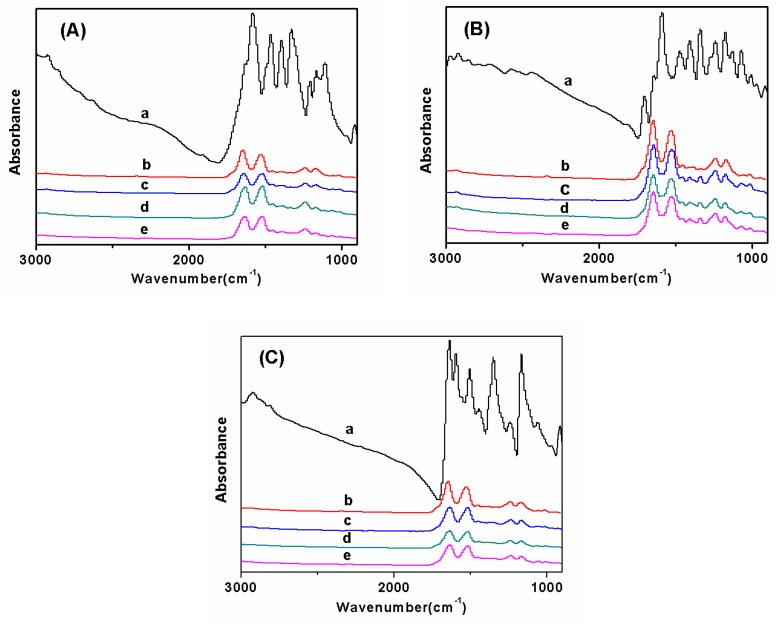
FTIR spectra of fluorescent SF nanofibers with different fluorescent dyes: (**A**) FS, (**B**) RB, and (**C**) AO; control sample: (**a**) fluorescent dyes; (**b**) pure SF nanofibers; the concentration of SF in electrospinning solution with 1.0 wt% fluorescent dyes was (**c**) 6.0 wt%, (**d**) 8.0 wt%, and (**e**) 10.0 wt%, respectively.

**Figure 5 polymers-11-00986-f005:**
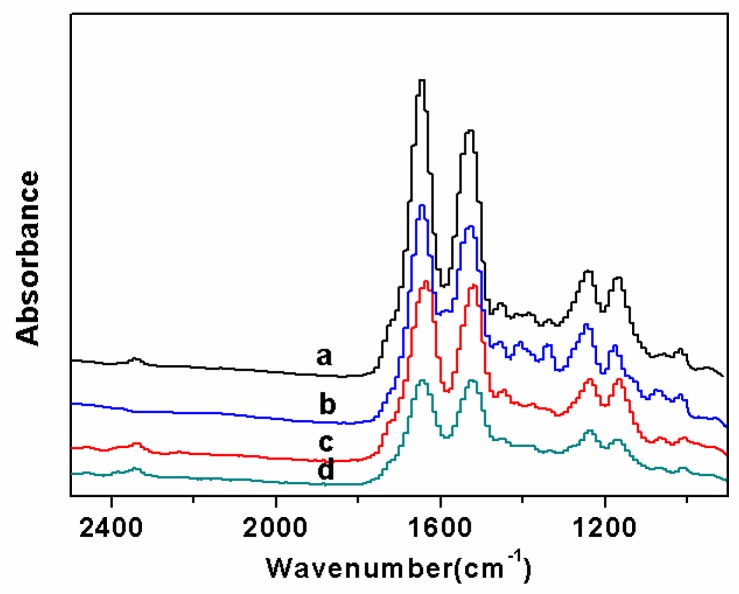
FTIR spectra of fluorescent SF nanofibers with different concentration of fluorescent sodium: (**a**) 0 wt%, (**b**) 0.5 wt%, (**c**) 1.0 wt%, and (**d**) 2.0 wt%, respectively. At the same time, the concentration of SF in electrospinning solution was 10.0 wt%.

**Figure 6 polymers-11-00986-f006:**
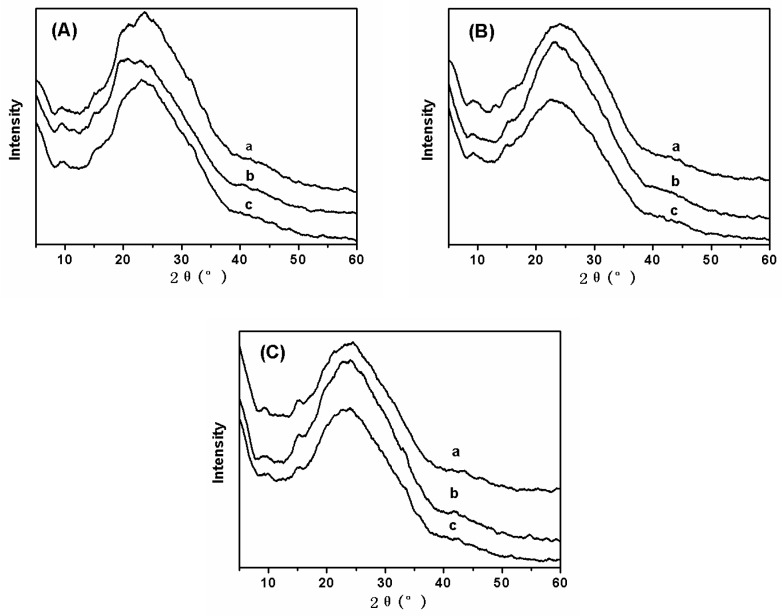
XRD data of fluorescent SF nanofibers with different fluorescent dyes: (**A**) FS, (**B**) RB, and (**C**) AO. At the same time, the concentrations of fluorescent dyes in electrospinning solution were (**a**) 0.5 wt%, (**b**) 1.0 wt%, and (**c**) 2.0 wt%, respectively. The concentration of SF in electrospinning solution was 10.0 wt%.

**Figure 7 polymers-11-00986-f007:**
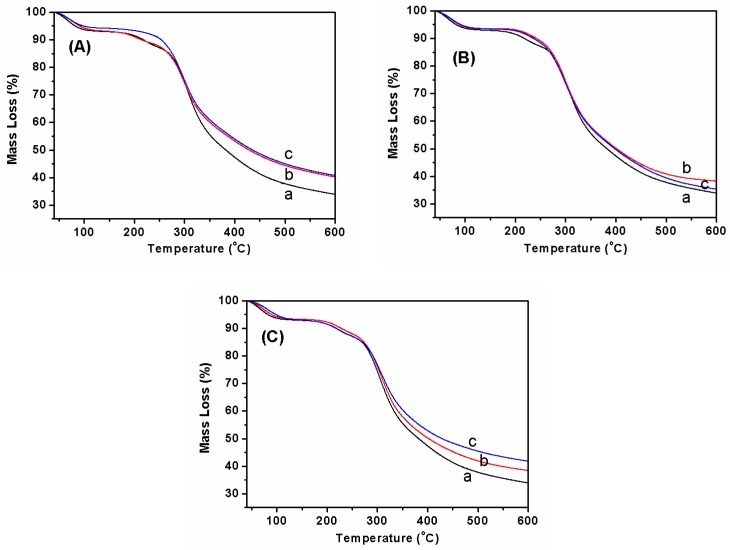
Thermogravimetric curves of fluorescent nanofibers influenced by different types and contents of fluorescent dyes. The type of fluorescent dye was (**A**) FS, (**B**) RB, and (**C**) AO, respectively. At the same time, the concentration of fluorescent dye was (**a**) 0, (**b**) 1.0 wt%, and (**c**) 2.0 wt%, respectively.

**Figure 8 polymers-11-00986-f008:**
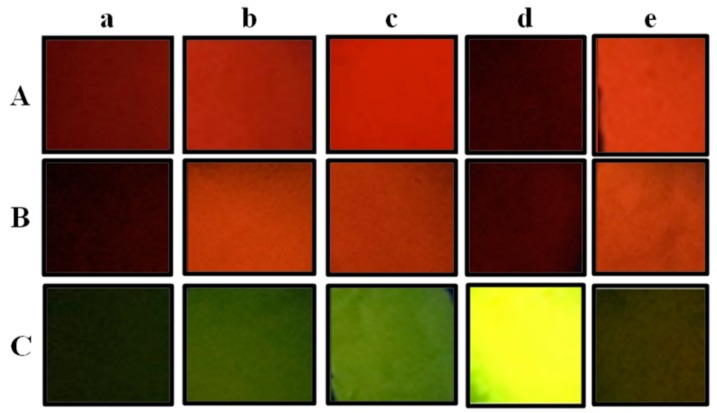
Fluorescence effect of fluorescent SF nanofibers irradiated by UV light: (**A**) RB, (**B**) AO, and (**C**) FS. The concentration of fluorescent dyes was (**a**–**c**) 1.0 wt%, (**d**) 0.5 wt%, and (**e**) 2.0 wt%. The concentration of SF was (**a**) 6.0 wt%, (**b**) 8.0 wt%, and (**c**–**e**) 10.0 wt%.

**Figure 9 polymers-11-00986-f009:**
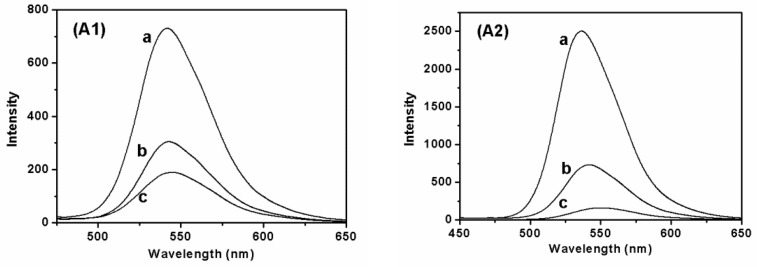
Fluorescence intensity of fluorescent SF nanofibers with different types of fluorescent dyes: (**A1**,**A2**) FS, (**B1**,**B2**) RB, (**C1**,**C2**) AO. (**A1**,**B1**,**C1**) The concentration of SF is (**a**) 10.0 wt%, (**b**) 8.0 wt%, and (**c**) 6.0 wt%. (**A2**,**B2**,**C2**) The concentration of fluorescent dyes is (**a**) 0.5 wt%, (**b**) 1.0 wt%, and (**c**) 2.0 wt%.

**Table 1 polymers-11-00986-t001:** Ratios of SF and fluorescent agents in spinning solution.

Materials	Concentration				
SF/wt%	6.0	8.0	10.0	10.0	10.0
Rhodamine B/wt%	1.0	1.0	1.0	0.5	2.0
Acridine orange/wt%	1.0	1.0	1.0	0.5	2.0
Fluorescent sodium/wt%	1.0	1.0	1.0	0.5	2.0

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
