# Peer review of "Fabrication of Silk Fibroin Fluorescent Nanofibers via Electrospinning"

_polymers, 2019, doi:10.3390/polym11060986_

Round 1

Reviewer 1 Report

In this manuscript, the authors reported a routine work about fabricating electrospun fluorescent silk fibroin nanofibers by adding three different dyes. Although the experiment is well designed, and the characterization is comprehensive, the novelty/significance of this work is very limited. Therefore, it is only commended for publication after all the comments/questions are addressed.

1. The supplementary figures are missing.

2. In Figure 1, the caption for "f" is missing.

3. How many fibers were calculated to get the average diameter by using imageJ? The size distribution is suggested to add.

4. What's the potential applications for using fluorescent SF nanofibers as a textile material? The author should provide more information/explanation.

5. As a potential textile material, the mechanical properties should be tested.

6. How is the fluorescent dye dispersed in the electrospun fibers? Inside the fiber, on the surface, or uniformly dispersed?

7. In the fluorescent performance analysis, fluorescence emission shift was observed. Please explain the reasons of this shift.

8. Please further address the novelty/significance of this work in the introduction section.

Author Response

1.The supplementary figures are missing.

Reply: The supplementary figures are added.

2. In Figure 1, the caption for "f" is missing.

Reply: The caption for “f” is added.

3. How many fibers were calculated to get the average diameter by using imageJ? The size distribution is suggested to add.

Reply: One hundred nanofibers were calculated to get the average diameter by using image J. The diameter distributing of nanofibers is added (supplementary parts).

4. What's the potential applications for using fluorescent SF nanofibers as a textile material? The author should provide more information/explanation.

Reply: the potential applications for using fluorescent SF nanofibers as a textile material are for fabric accessories such as fluorescent strip, fluorescent icon, etc.

5. As a potential textile material, the mechanical properties should be tested.

Reply: Fluorescent SF nanofibers are used for fabric accessories such as fluorescent strip, fluorescent icon, etc. The strength requirement of fluorescent nanofiber is not high. So, the mechanical properties of fluorescent nanofibers are further studied.

6. How is the fluorescent dye dispersed in the electrospun fibers? Inside the fiber, on the surface, or uniformly dispersed?

Reply: Fluorescent dyes are added in spinning solution, and then nanofibers are obtained through electrospinning method. So, I think fluorescent dyes are uniformly dispersed in nanofibers.

7. In the fluorescent performance analysis, fluorescence emission shift was observed. Please explain the reasons of this shift.

Reply: The reasons of this shift are added more explanation.

8. Please further address the novelty/significance of this work in the introduction section.

Reply: The introduction part is further modified.

Reviewer 2 Report

See comments in the attached file.

Author Response

1. See comment to this point in the conclusion section. (These fluorescent nanofibers may make it a promising textile materials for large scale application.)

Reply: These fluorescent nanofibers may make it promising textile materials for large scale application.

2. I imagine that FA refers to Formic Acid. But, please, define the acronym. (2.2 part)

Reply: Formic acid (FA)

3. Supplementary Figs. are not given.

Reply: The supplementary figures are added.

4. Please, provide the software version used.

Reply: The software version is SPSS14.0.

5. What was the concentration of pure SF? (Pure SF nanofibers exhibited a circular cross-section with a smooth surface (Fig.1f))

Reply: The concentration of pure SF is 6.0 wt% (Fig.1f).

6. P131, SF solution concentration? (3.1 part)

Reply: The sentence “with increasing SF concentration…” is revised to “with increasing SF solution concentration…”.

7. At the concentration of SF 10.0 wt%, the average diameter was 1200 for 0.5 wt% FS, 683 for 1 wt%, and 873 for 2 wt% FS. So, the fiber diameter decreases from 0.5 wt% FS to 1%, and then increases for 2% FS. Authors should comment about this behavior of fiber diameter.

Reply: The comments for fiber diameter changes are added.

8. Again there is an odd behavior in the fiber diameter with the content of RB: the diameter decreases from 0.5 wt% RB to 1 wt% RB, and then increases from 1 wt% RB to 2 wt%RB. Authors should comment about this behavior of fiber diameter.

Reply: The comments are added.

9. There is an odd behavior in the fiber diameter with the content of SF. Authors should comment about this behavior of fiber diameter. (Fig.3)

Reply: The comments are added.

10. Once again, there is an odd behavior in the fiber diameter with the content of AO. Authors should comment about this behavior of fiber diameter.

Reply: The comments are added.

11. In Figs 1, 2 and 3, it could be interesting to show also the fiber diameter distribution, not only the average and standard deviation values.

Reply: The fiber diameter distribution is added (supplementary parts).

12. This is not so clear: it depends on the concentration that you compare: for instances if you compare Figs. 1a, 2a, and 3a or, Figs. 1b, 2b, and 3b, or Figs. 1d, 2d, and 3d, the differences are larger: there is a stronger influence of the fluorescent dyes in the fiber morphology.

Reply: The purpose of this comparison is to analyze the effect of different fluorescent dyes on the morphology and structure of fibers.

13. This sentence is not correct: the concentration of SF was in the electrospinning solution, not in the nanofibers. Please, re-phrase this sentence. This error appears in other paragraphs of the paper. Please, revise all of them. (Fig.4)

Reply: The errors are revised.

14. What are the units for the intensity?

Figure A2: please, use the same color code for a, b and c than in figures B2 and C2.

Reply: The color code for figures is revised.

15. I have not clear the possible applications of this material. The authors should give more insight on this point. They have to explain why this material is promising for textile applications, compare properties with other articles in literature for this application, ...

In particular, I think that the authors have not given the mechanical properties of the material, that it should be a very important point for textile applications.

Reply: The potential applications for using fluorescent SF nanofibers as a textile material are for fabric accessories such as fluorescent strip, fluorescent icon, etc. For fabric accessories application, the strength requirement of fluorescent nanofiber is not high. So, the mechanical properties of fluorescent nanofibers are further studied.

Round 2

Reviewer 1 Report

The authors addressed some of the comments. However, (1) in comment 5, the authors replied that "the mechanical properties of fluorescent nanofibers are further studied", but I can’t find the mechanical properties data. (2) In comment 7, I can’t find the explanation in the revised manuscript. Please add and highlight it. (3) In comment 8, I can’t find the further modification in the introduction. Please add and highlight it.

Author Response

Reviewer #1

1. As a potential textile material, the mechanical properties should be tested.

Reply: Fluorescent SF nanofibers are used for fabric accessories such as fluorescent strip, fluorescent icon, etc. The strength requirement of fluorescent nanofiber is not high. The mechanical properties of fluorescent nanofibers are not significant influence factor for fabric accessories application. So, the results of mechanical properties are not provided in this paper.

2. In the fluorescent performance analysis, fluorescence emission shift was observed. Please explain the reasons of this shift.

Reply: The reasons of this shift are added more explanation. (The fluorescence emission shifted was correlated to the fluorescence effect of fluorescent SF nanofibers irradiated by UV light (Fig.8). The fluorescence effect of fluorescent SF nanofibers was changed by changing the concentration of FS. Hence, the fluorescence emission shifted was also affected by the concentration variations of FS.) (This phenomenon for fluorescence emission shift change was also attributed to the fluorescence effect of nanofibers, which were affected by the concentration of RB and AO, respectively.)

3. Please further address the novelty/significance of this work in the introduction section.

Reply: The introduction part is further modified. (There have been several reports on the biomedical and biotechnological applications using fluorescent silk fibroin. However, there is almost no application in the field of textiles such as fluorescent strip, fluorescent icon for apparels. In this paper, we report herein the demonstration of electrospun fluorescent silk nanofibers with stable fluorescence performance for textile application. )

Round 3

Reviewer 1 Report

The authors addressed all my concerns. The manuscript is recommended to publish.

Author Response

Reviewer #1

1. English language and style are fine/minor spell check required.

Reply: This paper is carefully checked, and minor spells are revised.

2.2 part, the sentence “the concentration of SF and fluorescent agents in FA solution was …” is revised to “the concentration of SF and fluorescent agents in FA solution were …”

3.1 part, the sentence “Pure SF nanofibers exhibited a circular cross-section with a smooth surface” is revised to “Pure SF nanofibers exhibited circular cross-section with smooth surface”.

The sentence “electrospun SF-RB nanofibers have smooth surface and its diameters was” is revised to “electrospun SF-RB nanofibers have smooth surface and its diameters were”.

The sentence “electrospun SF-AO nanofibers have smooth surface and its diameters was” is revised to “electrospun SF-AO nanofibers have smooth surface and its diameters were”.

3.2 part, the sentence “The peaks of pure SF nanofibers sample was characterized” is revised to “The peaks of pure SF nanofibers were characterized”.

The sentence “It also showed different fluorescent dyes was” is revised to “It also showed different fluorescent dyes were”.
